

# A numbering algorithm for finite elements on extruded meshes which avoids the unstructured mesh penalty

Gheorghe-Teodor Bercea[1], Andrew T. T. McRae[2,3,4], David A. Ham[3], Lawrence Mitchell[1],
Florian Rathgeber[1,5], Luigi Nardi[1], Fabio Luporini[1], and Paul H. J. Kelly[1]

[1]Department of Computing, Imperial College London, London, SW7 2AZ, United Kingdom
[2]The Grantham Institute, Imperial College London, London, SW7 2AZ, United Kingdom
[3]Department of Mathematics, Imperial College London, London, SW7 2AZ, United Kingdom
[4]Department of Mathematical Sciences, University of Bath, Bath, BA2 7AY, United Kingdom
[5]European Centre for Medium-Range Weather Forecasts (ECMWF), Reading, RG2 9AX, United Kingdom

*Correspondence to:* Gheorghe-Teodor Bercea (gb308@doc.ic.ac.uk)

**Abstract.** We present a generic algorithm for numbering and then efficiently iterating over the data values attached to an extruded mesh. An extruded mesh is formed by replicating an existing mesh, assumed to be unstructured, to form layers of prismatic cells. Applications of extruded meshes include, but are not limited to, the representation of 3D high aspect ratio domains employed by geophysical finite element simulations. These meshes are structured in the extruded direction. The algorithm presented here exploits this structure to avoid the performance penalty traditionally associated with unstructured meshes. We evaluate our algorithm on a range of low compute intensity operations which constitute worst cases for data layout performance exploration. The experiments show that having structure along the extruded direction enables the cost of the indirect data accesses to be amortized after 10-20 layers as long as the underlying mesh is well-ordered. We characterise the resulting spatial and temporal reuse in a representative set of both continuous-Galerkin and discontinuous-Galerkin discretisations. On meshes with realistic numbers of layers the performance achieved is between 70% and 90% of a theoretical hardware-specific limit.

*Keywords:* extruded meshes, code generation, finite elements, locality optimisation

## 1 Introduction

In the field of numerical simulation of fluids and structures, there is traditionally considered to be a tension between the computational efficiency and ease of implementation of structured grid models, and the flexible geometry and resolution offered by unstructured meshes.

In particular, one of the grand challenges in simulation science is modelling the ocean and atmosphere for the purposes of predicting the weather or understanding the Earth's climate system. The current generation of large-scale operational atmosphere and ocean models almost all employ structured meshes (Slingo et al., 2009). However, requirements for geometric flexibility as well as the need to overcome scalability issues created by the poles of structured meshes has led in recent years



to a number of national projects to create unstructured mesh models (Ford et al., 2013; Zängl et al., 2015; Skamarock et al., 2012).

The ocean and atmosphere are thin shells on the Earth's surface, with typical domain aspect ratios in the thousands (oceans are a few kilometres deep but thousands of kilometres across). Additionally the direction of gravity and the stratification of the ocean and atmosphere create important scale separations between the vertical and horizontal directions. The consequence of this is that even unstructured mesh models of the ocean and atmosphere are in fact only unstructured in the horizontal direction, while the mesh is composed of aligned layers in the vertical direction. In other words, the meshes employed in the new generation of models are the result of extruding an unstructured two-dimensional mesh to form a layered mesh of prismatic elements.

This layered structure was exploited in Macdonald et al. (2011) to create a numbering for a finite volume atmospheric model such that iteration from one cell to the next within a vertical column required only direct addressing. They show that when only paying the price of indirect addressing on the base mesh there is less than 5% performance difference between two implementations of an atmospheric model which treat the same icosahedral mesh first as fully structured and then as partially structured (extruded). One of the caveats of that comparison is that the underlying mesh is fully structured in both cases which presents an advantage to the indirect addressing scheme which is not present for more general unstructured meshes.

A key motivation for this work was to provide an efficient mechanism for the implementation of the layered finite element numerics which have been adopted by the UK Met Office's Gung Ho programme to develop a new atmospheric dynamical core. The algorithms here have been adopted by the Met Office for this purpose (Ford et al., 2013). While geophysical applications motivate this work, the algorithms and their implementation in Firedrake (Rathgeber et al., 2015) are more general and could be applied to any high aspect ratio domain.

## 1.1 Contributions

- We generalize the numbering algorithm in Macdonald et al. (2011) to the full range of finite element discretisations.

- We demonstrate the effectiveness of the algorithm with respect to absolute hardware performance limits.

## 2 Unstructured Meshes

In this section we briefly restate the data model for unstructured meshes introduced in Logg (2009); Knepley and Karpeev (2009). In Section 2.2 we rigorously define a *mesh*, explain mesh topology, geometry and numbering. In Section 2.3 we explain how data may be associated with meshes.

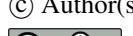



## 2.1 Terminology

When describing a mesh, we need some way of specifying the neighbours of a given entity. This is always possible using *indirect addressing* in which the neighbours are explicitly enumerated, and sometimes possible with *direct addressing* where a closed form mathematical expression suffices.

In what follows we start with a *base mesh* which we will *extrude* to form a mesh of higher topological dimension. Due to geophysical considerations, we refer to the plane of the base mesh as the *horizontal* and to the layers as the *vertical*.

We will also employ the definition of a *graph* as a set $V$ and a set $E$ of edges where each edge represents the relationships between the elements of the set $V$.

## 2.2 Meshes

A mesh is a decomposition of a simulation domain into non-overlapping polygonal or polyhedral cells. We consider meshes used in algorithms for the automatic numerical solution of partial differential equations. These meshes combine topology and geometry. The topology of a mesh is composed of mesh entities (such as vertices, edges, cells) and the adjacency relationships between them (cells to vertices or edges to cells). The geometry of the mesh is represented by coordinates which define the position of the mesh entities in space.

Every mesh entity has a topological dimension given by the minimum number of spatial dimensions required to represent that entity. We define $D$ to be the minimum number of spatial dimensions needed to represent a mesh and all its entities. A vertex is representable in zero-dimensional space, similarly an edge is a one-dimensional entity and a cell a $D$-dimensional entity. In a two-dimensional mesh of triangles, for example, the entities are the vertices, edges and triangle cells with topological dimensions 0, 1 and 2 respectively. The minimum number of geometric dimensions needed to represent the mesh and all its

entities is $D = 2$.

A mesh can be represented by several graphs. Each graph consists of a multi-type set $V$ and a typed adjacency relationship $\text{Adj}_{d_1,d_2}$ between $d_1$- and $d_2$-typed elements in $V$. The type of an entity in $V$ is simply its dimension. The adjacency graphs will always map from a set of uniform dimension to a set of uniform dimension. Attaching types to elements of $V$ enables graphs to capture the relationships between different mesh entities, for example cells and vertices, edges and vertices.

We write $V_d$ to mean the set of mesh entities of topological dimension $d$ where $0 \leq d \leq D$:

$$V_d = \{(d,i) \mid 0 \leq i \leq N_d - 1\}, \tag{1}$$

where $N_d$ is the number of entities of dimension $d$. The set $V$ is then simply the union of the $V_d$s:

$$V = \bigcup_{0 \leq d \leq D} V_d. \tag{2}$$

Every mesh entity has a number of adjacent entities. The mesh-element connectivity relationships are used to specify the

way mesh entities are connected. For a given mesh of topological dimension $D$ there are $(D+1)^2$ different types of adjacency relationships. To define the mesh, only a minimal subset of relationships from which all the others can be derived is required.



For example, as shown in Logg (2009), the complete set of adjacency relationships may be derived from the cell-vertex adjacency.

We write

$$\mathrm{Adj}_{d_1,d_2}(v) = (v_1, v_2, \ldots, v_k), \tag{3}$$

to specify the entities $v_1, v_2, \ldots, v_k \in V_{d_2}$ adjacent to $v \in V_{d_1}$.

In a mesh with a very regular topology, there may be a closed form mathematical expression for the adjacency relationship $\mathrm{Adj}_{d_1,d_2}(v)$. Such meshes are termed *structured*. However since we are also interested in supporting more general *unstructured meshes*, we must store the lists of adjacent entities explicitly.

### 2.3 Attaching data to meshes

Every mesh entity has a number of values associated with it. These values are also known as *degrees of freedom* and they are the discrete representation of the continuous data fields of the domain. As the degrees of freedom are uniquely associated with mesh entities, the mesh topology can be used to access the degrees of freedom local to any entity using the connectivity relationships.

A *finite element discretisation* associates a number of degrees of freedom with each entity of the mesh. A *function space* 15 uses the discretisation to define a numbering for all the degrees of freedom. Multiple different function spaces may be defined on a mesh and each function space may have several data fields associated with it. In the case of a triangular mesh for example, a piecewise linear function space will associate a degree of freedom with every vertex of the mesh while a cubic function space will associate one degree of freedom with every vertex, two degrees of freedom with every edge and one degree of freedom with every cell. In the former case there will be three degrees of freedom adjacent to a cell, and a total of ten in the latter case.

The data associated with the mesh also needs to be numbered. The choice of numbering can have a significant effect on the computational efficiency of calculations over the mesh (Gunther et al., 2006; Lange et al., 2015; Yoon et al., 2005).

The most common operation performed on meshes is the local application of a function or *kernel* while traversing, or *iterating* over a homogeneous subset of mesh entities. When iterating over a specific mesh entity type, the kernel often consists of a stencil-like operation accessing nearby degrees of freedom. For example, in a finite element simulation over a triangle 25 mesh, when iterating over a cell, the kernel might require the degrees of freedom on the vertices, edges and the interior of the triangle. In theory, this requires cell-to-edges and cell-to-vertices adjacency relationships (cell-to-cell is implicit). In practice the three different relationships may be composed into a single adjacency relationship which references the data associated with all the different adjacent entity types.

In the unstructured case, we store an explicit list (also known as *map*) $L(e)$ for each type of stencil operation which given a 30 topological entity $e$ returns the set of degrees of freedom in the stencil at that entity.

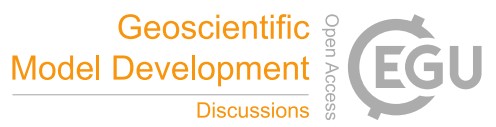

## 3 Extruded Meshes

In Section 3.1 we introduce extruded meshes and in Section 3.2 we show how the entities and the data are to be numbered. In Section 3.3 we present the extruded mesh iteration algorithm and the offset computation for the direct addressing scheme along the vertical direction.

### 3.1 Definition of an Extruded Mesh

An extruded mesh consists of a base mesh which is replicated a fixed number of times in a layered structure[1]. A mesh of topological dimension $D$ becomes an extruded mesh of topological dimension $D + 1$.

The mesh definition can be extended to include extruded meshes. Let mesh $M = (V, \mathrm{Adj})$ be a non-extruded mesh where $\mathrm{Adj}$ stands for all the valid adjacency relationships of $M$. An extruded mesh which has $M$ as the base mesh can be defined as a triple $(V^{\mathrm{extr}}, \mathrm{Adj}^{\mathrm{extr}}, \lambda)$ where $\mathrm{Adj}^{\mathrm{extr}}$ is the set of valid adjacency relationships and $\lambda \in \mathbb{N}^+$ is the number of layers of the extruded mesh. Before we can define $V^{\mathrm{extr}}$ and $\mathrm{Adj}^{\mathrm{extr}}$ several concepts have to be introduced.

#### 3.1.1 Tensor product cells

The effect of the extrusion process on the base mesh can always be captured by associating a line segment with the vertical direction. We write $D_b$ for the topological dimension of the base mesh while the topological dimension of the vertical mesh is always equal to $1$.

As a consequence, the cells of the extruded mesh are prisms formed by taking the tensor product of the base mesh cell with the vertical line segment. For example, each triangle becoems a triangular prism. The construction of tensor product cells and finite element spaces on them is considered in more detail in McRae et al. (2016).

#### 3.1.2 Extruded Mesh Entities

The extrusion process introduces new types of mesh entities reflecting the connectivity between layers. The pairs of corresponding entities of dimension $d$ in adjacent layers are connected using entities of dimension $d + 1$. In a triangular mesh for example (Fig. 1), the corresponding vertices are connected using vertical edges, edges contained in each layer are connected by quadrilateral facets and the 2D triangle faces are connected by a 3D triangular prism (Fig. 2).

The topological dimension on its own is no longer enough to distinguish between the different types of entities and their orientation. Instead entities are characterised by a pair composed of the horizontal and vertical dimensions. In the case of a 2D triangular base mesh the set of dimensions is $\{0, 1, 2\}$. The line segment of the vertical can be described by the set of dimensions $\{0, 1\}$. The Cartesian product of the two sets yields a set of pairs (4) which can be used to uniquely identify mesh entities.

$$\{(0,0), (0,1), (1,0), (1,1), (2,0), (2,1)\} \tag{4}$$

---

[1]For ease of exposition, we discuss the case where each mesh column contains the same number of layers, however this is not a limitation of the method and algorithms presented here

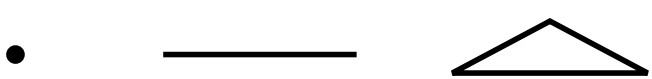

**Figure 1.** Extruded mesh entities belonging to the base mesh to be extruded (left to right): vertices, horizontal edges, horizontal facets.

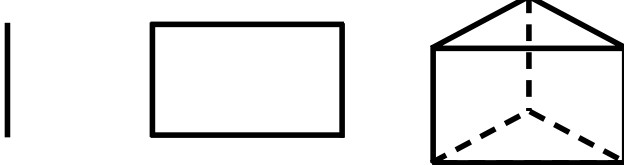

**Figure 2.** Mesh entities used in the extrusion process to connect entities in Fig. 1 (left to right): vertical edges, vertical facets, 3D cells.

We refer to the components of each pair as the *horizontal* and *vertical* dimension of the entity respectively. Table 1 shows the mapping between the mesh entity types and their descriptor.

### 3.1.3 Extruded Mesh Entity Numbering

We write $V_{d_1,d_2}$ to denote the set of topological entities which are the tensor product of entities of dimensions $d_1$ in the horizontal and $d_2$ in the vertical ($0 \leq d_1 \leq D_b$ and $0 \leq d_2 \leq 1$):

$$V_{d_1,d_2} = \{((d_1,d_2),(i,l)) \mid 0 \leq i \leq N_{d_1} - 1,\ 0 \leq l \leq \lambda - d_2\}, \tag{5}$$

where $N_{d_1}$ is the number of entities of dimension $d_1$ in the base mesh and $\lambda$ is the number of extruded layers. The subtraction of $d_2$ from the number of layers accounts for the fencepost error caused by the fact that there is always one fewer edge than vertex in the vertical direction.

The complete set of extruded mesh entities is then

$$V^{\text{extr}} = \bigcup_{\substack{0 \leq d_1 \leq D_b \\ 0 \leq d_2 \leq 1}} V_{d_1,d_2}. \tag{6}$$

Similarly we must extend the indexing of the adjacency relationships, writing:

$$\text{Adj}^{\text{extr}}_{(d_1,d_2),(d_3,d_4)}(v) = (v_1, v_2, \ldots, v_k), \tag{7}$$

where $v \in V_{d_1,d_2}$ and $v_1, v_2, \ldots, v_k \in V_{d_3,d_4}$.

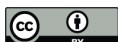



## 3.2 Attaching data to extruded meshes

Identically to the case of non-extruded meshes, function spaces over an extruded mesh associate degrees of freedom with the (extended) set of mesh entities. A constant number of degrees of freedom is associated with each entity of a given type.

If we can arrange that the degrees of freedom are numbered such that the vertical entities are "innermost", it is possible to use direct addressing for the vertical part of any mesh iteration, significantly reducing the computational penalty introduced by using an indirectly addressed, unstructured base mesh. Algorithm 1 implements this "vertical innermost" numbering algorithm. The critical feature of this algorithm is that degrees of freedom associated with vertically adjacent entities have adjacent global numbers. The outcome of this vertical numbering is shown in Fig. 3. The global numbering algorithm is orthogonal to any base mesh decomposition strategy used to support execution on distributed memory parallel systems.

---

**Algorithm 1** Computing the global numbering for degrees of freedom on an extruded mesh

---

**Input:** $V$ : the set of base mesh entities

**Input:** $\lambda$ : the number of layers

**Input:** $\delta((d_1, d_2))$ : the number of DoFs associated with each $(d_1, d_2)$ entity

**Output:** $\mathsf{dofs}_{\mathsf{fs}}$: the degrees of freedom associated with each entity

$\quad c \leftarrow 0$ {Loop over base mesh entities}

$\quad$ **for** $(d_1, i)$ in $V$ **do**

$\quad\quad$ {Loop over layers}

$\quad\quad$ **for** $l$ in $\{0, 1, ..., \lambda - 1\}$ **do**

$\quad\quad\quad$ {Number the horizontal layer, then the connecting entity above it}

$\quad\quad\quad$ **for** $d_2$ in $\{0, 1\}$ **do**

$\quad\quad\quad\quad$ {Assign the next $\delta((d_1, d_2))$ global DoF numbers to this entity}

$\quad\quad\quad\quad \mathsf{dofs}_{\mathsf{fs}}((d_1, d_2), (i, l)) \leftarrow c, c + 1, ..., c + \delta((d_1, d_2)) - 1$

$\quad\quad\quad\quad c \leftarrow c + \delta((d_1, d_2))$

$\quad\quad\quad$ **end for**

$\quad\quad$ **end for**

$\quad\quad$ {Number the top horizontal layer of this column}

$\quad\quad \mathsf{dofs}_{\mathsf{fs}}[(d_1, 0), (i, l)] \leftarrow c, c + 1, ..., c + \delta((d_1, 0)) - 1$

$\quad\quad c \leftarrow c + \delta((d_1, 0))$

$\quad$ **end for**

---

## 3.3 Iterating over extruded meshes

Iterating over the mesh and applying a kernel to a set of connected entities (stencil) is the key operation used in mesh-based computations.

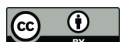

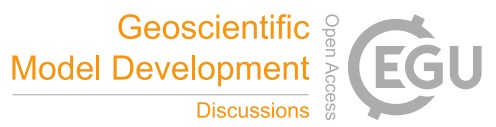

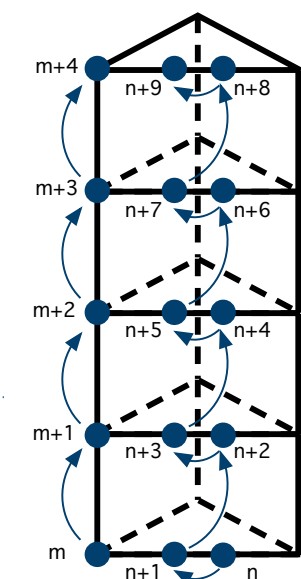

**Figure 3.** Vertical numbering of degrees of freedom (shown in filled circles) associated with vertices and horizontal edges. Only one set of vertically aligned degrees of freedom of each type is shown. The arrows outline the order in which the degrees of freedom are numbered.

The global numbering of the degrees of freedom allows stencils to be calculated using a direct addressing scheme when accessing the degrees of freedom of vertically adjacent entities. We assume that the traversal of the mesh occurs over a set of mesh entities which is homogeneous (a set containing only cells for example). Degrees of freedom belonging to vertically adjacent entities, accessed by two consecutive kernel applications on the same column, have a constant offset between them.

5    The offset is given by the sum of degrees of freedom attached to the two vertically adjacent entities contained in the stencil:

$$\delta((d,0)) + \delta((d,1)) \tag{8}$$

Let $S$ be the stencil of a kernel which needs to access the values of the degrees of freedom of a field $f$ defined on a function space fs. Let $L_{\mathsf{fs}}(v) = (\mathsf{dof}_0, \mathsf{dof}_1, ..., \mathsf{dof}_{k-1})$ be the list of degrees of freedom of the stencil for an input entity $v \in V_{d_1, d_2}$.

The lists of degrees of freedom accessed by $S$ could be provided explicitly for all the input entities $v$. Using the previous

10    result we can instead reduce the number of explicitly provided lists by a factor of $\lambda$. For each column we visit, the only explicit accesses required are the ones to the degrees of freedom at the bottom of the column. The degrees of freedom identifiers for the rest of the stencil applications in the same column can be obtained by adding a multiple of the constant vertical offset to each degree of freedom in the bottom explicit list.

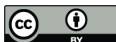
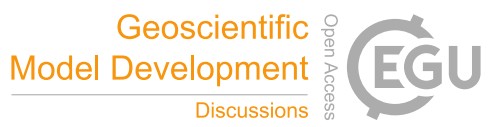

For a given stencil function $S$ an offset can be computed for each degree of freedom in the corresponding explicit list $L_{\mathsf{fs}}$. As the ordering of the degrees of freedom in the stencil is fixed (by consistent ordering of mesh entities) the vertical offset only needs to be computed once for a particular function space $fs$.

The algorithm for computing the vertical offset is presented in Algorithm 2. Note that since the offset for two vertically aligned entity types is the same, only the base mesh entity type is considered.

---

**Algorithm 2** Computation of vertical offsets

---

**Input:** $k$ : number of degrees of freedom accessed by stencil function $S$

**Input:** $E_S(i)$: the base mesh entity type of the $i$-th degree of freedom accessed by $S$

**Output:** $\mathsf{offset}_{S,\mathsf{fs}}$ : the vertical offset for function space fs given stencil $S$

    **for** $i$ in $\{0, 1, ..., k-1\}$ **do**

        $d \leftarrow E_S(i)$

        $\mathsf{offset}_{S,\mathsf{fs}}(i) \leftarrow \delta((d,0)) + \delta((d,1))$

    **end for**

---

If $(\mathsf{dof}_0, \mathsf{dof}_1, ..., \mathsf{dof}_{k-1})$ is the explicit list of degrees of freedom for the initial layer to which the stencil can be applied, then the list of degrees of freedom for the $n^{\mathrm{th}}$ application of the stencil along the vertical ($n < \lambda$) is given by:

$$(\mathsf{dof}_0 + n \times (\mathsf{offset}_{S,f}(0)), ..., \mathsf{dof}_{k-1} + n \times (\mathsf{offset}_{S,f}(k-1))) \tag{9}$$

Algorithm 3 shows the iteration algorithm working for a single field $f$ on a function space fs. The stencil function $S$ is applied to the entities of each column in turn. Each time the algorithm moves on to the next vertically adjacent entity, the indices of the degrees of freedom accessed are incremented by the vertical offset $\mathsf{offset}_{S,\mathsf{fs}}$. The algorithm is also applicable to stencil functions of multiple fields defined on the same function space since the data associated with each field is accessible using the same set of degree of freedom numbers. The extension to fields from different function spaces just requires explicit lists $L_{\mathsf{fs}}$ for each space.

## 4 Performance Evaluation

In this section, we test the hypothesis that iteration exploiting the extruded structure of the mesh amortizes the unstructured base mesh overhead of accessing memory through explicit neighbour lists. We also show that the more layers the mesh contains, the closer its performance is to the hardware limits of the machine.

We validate our hypotheses in the Firedrake finite element framework (Rathgeber et al., 2015). Although we restrict our performance evaluation to examples drawn from finite element discretisations, the algorithms we have presented can be applied to any mesh-based discretisation.

In Section 4.1 we describe the design of the experiments undertaken. The hardware platforms and the methodology used are described in Section 4.2 followed by results and discussion in Sections 4.3 and 4.4 respectively.

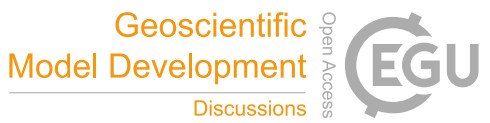

---

**Algorithm 3** Iteration of a stencil function over an extruded mesh

---

**Input:** $V$: iteration set of base mesh entities

**Input:** $S$: stencil function to be applied to the degrees of freedom of field $f$

**Input:** $L_{\mathsf{fs}}$: set of explicit lists of degrees of freedom for function space $\mathsf{fs}$

**Input:** $\mathsf{offset}_{S,\mathsf{fs}}$: the vertical offset for function space $\mathsf{fs}$ given stencil $S$

  **for** $v$ in $V$ **do**

    $(\mathsf{dof}_0, \mathsf{dof}_1, ..., \mathsf{dof}_{k-1}) \leftarrow L(v)$

    **for** $l$ in $\{0, 1, ..., \lambda - d_2\}$ **do**

      $S(f(\mathsf{dof}_0), f(\mathsf{dof}_1), ..., f(\mathsf{dof}_{k-1}))$

      **for** $j$ in $\{0, 1, ..., k - 1\}$ **do**

        $\mathsf{dof}_j \leftarrow \mathsf{dof}_j + \mathsf{offset}_{S,\mathsf{fs}}(j)$

      **end for**

    **end for**

  **end for**

---

## 4.1 Experimental Design

The design space to be explored is parameterized by number of layers and the manner in which the data is associated with the mesh and therefore accessed. In establishing the relationship between the performance and the hardware we examine performance on two generations of processors and varying process counts.

### 4.1.1 Choosing the computation

Numerical computations of integrals are the core mesh iteration operation in the finite element method. We focus on residual (vector) assembly for two reasons. First, in contrast to Jacobian assembly, there are no overheads due to sparse matrix insertion; the experiment is purely a test of data access via the mesh indirections. Second, residual evaluation is the assembly operation with the lowest computational intensity and therefore constitutes a worst-case scenario for data layout performance exploration. Since we are interested in data accesses, we choose the simplest non-trivial residual assembly operation:

$$I_1 = \int_\Omega f v \, \mathrm{d}x, \quad \forall v \in V \tag{10}$$

for $f$ in the finite element space $V$.

In addition to the output field $I_1$ and the input field $f$ this computation accesses the coordinate field, $\boldsymbol{x}$. Regardless of the choice of $V$, we always represent $\boldsymbol{x}$ by a $d$-vector at each vertex of the $d$-dimensional mesh.

### 4.1.2 Choosing the discretisations

The construction of a wide variety of finite element spaces on extruded meshes was introduced in McRae et al. (2016). This enables us to select the horizontal and vertical data discretisations independently.



For the purposes of data access, the distinguishing feature of different finite element spaces is the extent to which degrees of freedom are shared between adjacent cells.

We choose a set of finite element spaces spanning the combinations of horizontal and vertical reuse patterns found on extruded meshes: horizontal and vertical reuse, only horizontal, only vertical, or no reuse at all.

5    We employ low order continuous and discontinuous discretisations (abbreviated as *CG* and *DG* respectively) in both the horizontal and vertical directions.

The set of discretisations is $A = \{\mathrm{CG1}, \mathrm{DG0}, \mathrm{DG1}\}$ where the number indicates the degree of polynomials in the space. We examine all pairs of discretisations $(h, v) \in A \times A$. Since the cells of the base mesh are triangles, the extruded mesh consists of triangular prisms. Fig. 4 shows the data layout of each of these finite elements.

10    Both Firedrake and our numbering algorithm support a much larger range of finite element spaces than this. However, the more complex and higher degree spaces will result in more computationally intensive kernels but not materially different data reuse. The lowest order spaces are the most severe test of our approach since they are more likely to be memory bound.

### 4.1.3   Layer count and problem size

We vary the number of layers between 1 and 100. This is a realistic range for current ocean and atmosphere simulations. 15    The number of cells in the extruded mesh is kept approximately constant by shrinking the base mesh as the number of layers increases. The mesh size is chosen such that the data volume far exceeds the total last level cache capacity of each chosen architecture (L3 cache in all cases). This minimizes caching benefits and is therefore the strongest test of our algorithms. The overall mesh size is fixed at approximately 15 million cells which yields a data volume of between 300 and 840 MB depending on discretisation.

### 4.1.4   Base mesh numbering

The order in which the entities of the unstructured mesh are numbered is known to be critical for data access performance. To characterize this effect and distinguish it from the impact of the number of layers, we employ two variants of each base mesh. The first is a mesh for which the traversal is optimised using a reverse Cuthill-McKee ordering (Lange et al., 2015). The second is a *badly* ordered mesh with a random numbering. This represents a pathological case for temporal locality.

### 4.2   Experimental Setup

The specification of the hardware used to conduct the experiments is shown in Table 2. Following Ofenbeck et al. (2014) we disable the Intel turbo boost and frequency scaling. This is intended to prevent our performance results from being subject to fluctuations due to processor temperature.

The experiments we are considering are run on a single two-socket machine and use MPI (Message Passing Interface) 30    parallelism. The number of MPI processes varies from one up to 2 processes per physical core (exploiting hyperthreading). We pin the processes evenly across physical cores to ensure load balance and prevent process migration between cores.



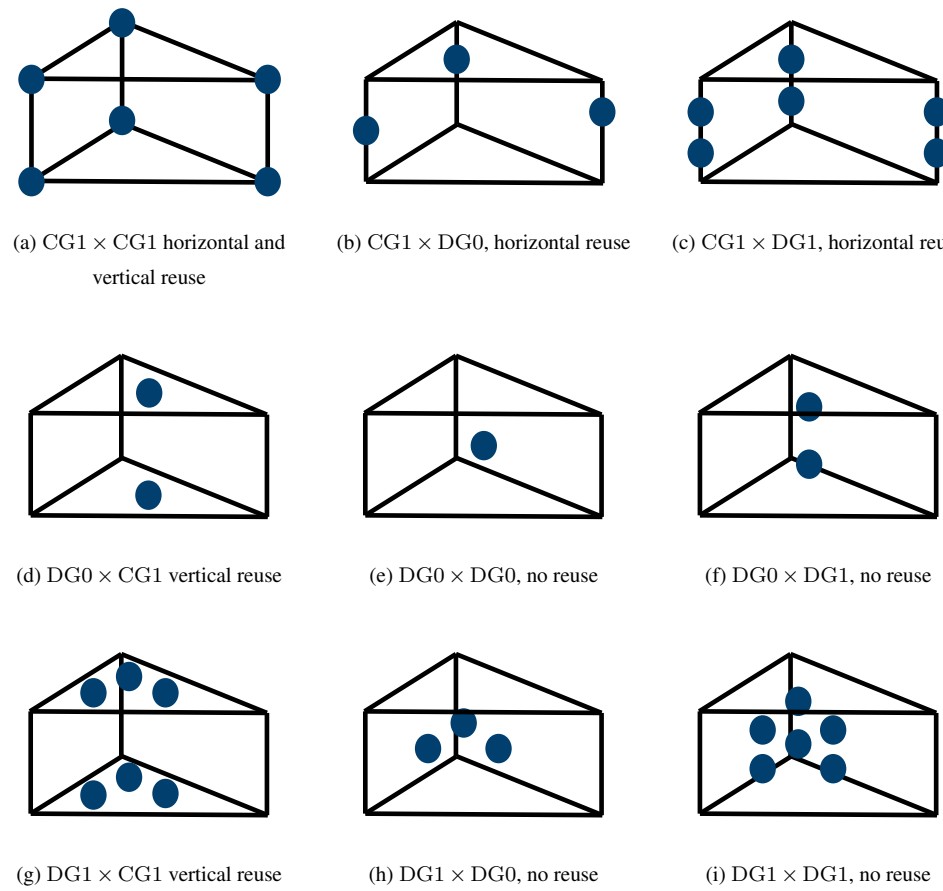

(a) CG1 × CG1 horizontal and vertical reuse

(b) CG1 × DG0, horizontal reuse

(c) CG1 × DG1, horizontal reuse

(d) DG0 × CG1 vertical reuse

(e) DG0 × DG0, no reuse

(f) DG0 × DG1, no reuse

(g) DG1 × CG1 vertical reuse

(h) DG1 × DG0, no reuse

(i) DG1 × DG1, no reuse

**Figure 4.** Tensor product finite elements with different data layout and cell-to-cell data re-use.

The Firedrake platform performs integral computations by automatically generating *C* code. The compiler used is GCC version 4.9.1 (`-O3 -march=native -ffast-math -fassociative-math`). We also assessed the performance of the Intel C Compiler version 15.0.2 (`-O3 -xAVX -ip -xHost`), however we only report results from GCC in this paper since the performance of the Intel compiler was inferior.

5  **4.2.1   Runtime, data volume, bandwidth and FLOPs**

Runtime is measured using a nanosecond precision timer. Each experiment is performed ten times and we report the minimum runtime. Exclusive access to the hardware has been ensured for all experiments.

Different discretisations lead to different data volumes due to the way data is shared between cells. DG based discretisations require the movement of larger data volumes while CG discretisations lead to smaller volumes due to data reuse. To evaluate
10  the impact of different data volumes we compare the valuable bandwidth with the achieved STREAM bandwidth.

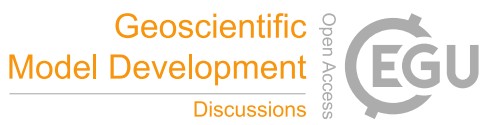

We model the data transfer from main memory to CPU assuming a perfect cache: each piece of data is only loaded from main memory once. We define the *valuable data volume* as the total size of the input, output and coordinate fields. This gives a lower bound on the memory traffic to and from main memory. The valuable data volume divided by the runtime yields the *valuable bandwidth*.

The maximum bandwidth achieved for the STREAM triad benchmark (McCalpin, 1995) is shown in Table 3. The percentage of STREAM bandwidth achieved by the valuable bandwidth shows how prone the code is to becoming bandwidth bound as the floating point performance of the payload is improved.

The floating point operations – adds, multiplies and, on Haswell, fused multiply-add (FMA) operations – are counted automatically using the Intel Architecture Code Analyzer (Intel, 2012) whose results are verified with PAPI (Mucci et al., 1999)
which accesses the hardware counters.

### 4.2.2 Theoretical performance bounds

The performance of the extruded iteration depends on the efficiency of the generated finite element kernel (payload) code which for some cases may not be vectorized (as outlined in Luporini et al. (2015)) or may not have a perfectly balanced number of floating point additions and multiplications. Kernel code optimality is outside the scope of this paper.

To a first approximation the performance of a numerical algorithm will be limited by either the memory bandwidth or the floating point throughput. The STREAM benchmark provides an effective upper bound on the achievable memory bandwidth. The floating point bounds employed are based on the theoretical maximum given the clock frequency of the processor.

The Intel architectures considered are capable of executing both a floating point addition and a floating point multiplication on each clock cycle. The Haswell processor can execute a fused multiply-add instruction (FMA) instead of either an addition
or multiplication operation.

The achievable FLOP rate may therefore be as much as twice the clock rate depending on the mix of instructions executed. The achievable speed-up over the clock rate, $f_b$, for the Sandy Bridge platform is therefore bounded by the balance factor

$$f_b = 1 + \frac{\min(\text{add FLOPs}, \text{multiplication FLOPs})}{\max(\text{add FLOPs}, \text{multiplication FLOPs})}, \tag{11}$$

while for Haswell it is bounded by

$$f_b = 1 + \frac{\min(\text{add FLOPs}, \text{multiplication FLOPs}) + k}{\max(\text{add FLOPs}, \text{multiplication FLOPs}) + k}, \tag{12}$$

where $k$ is half the number of FMAs.

### 4.2.3 Vectorization

The processors employed support 256-bit wide vector floating point instructions. The double precision FLOP rate of a fully vectorized code can be as much as four times the clock rate. GCC automatically vectorized only a part of the total number of
floating point instructions. The ratio between the number of vector (packed) floating point instructions and the total number of floating point instructions (scalar and packed) characterizes the impact of partial vectorization on the floating point bound

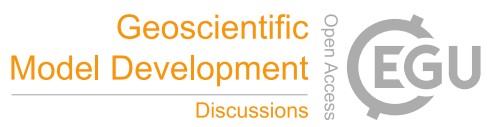

through the vectorization factor

$$f_v = 1 + (4-1) \times \frac{\text{vector FLOPs}}{\text{total FLOPs}}. \tag{13}$$

To control the impact of the kernel computation (payload) on the evaluation, we compare the measured floating point throughput with a theoretical peak which incorporates the payload instruction balance and the degree of vectorization. Let $c$ be the number of active CPU physical cores during the computation of interest. The base (theoretical) floating point performance $B_c$ is the same for all discretisations and assumes one floating point instruction per cycle for each (active) physical CPU core. The peak theoretical floating point throughput $P_d$ is different for each discretisation $d$ as it depends on the properties of the payload and is given by

$$P_d = B_c \times f_b \times f_v. \tag{14}$$

### 4.3 Experimental Results

#### 4.3.1 Amortizing the cost of indirect accesses

When the base mesh is well ordered (Fig. 6), the number of layers required to reach a performance plateau is between 10 and 20 for all discretisations. When the base mesh is badly ordered (Fig. 5) the required number of layers can be as large as fifty or more. For example, in the case of discretisations employing DG0 either horizontally or vertically, the FLOP rate plateau is not reached even at a hundred layers.

#### 4.3.2 Percentage of theoretical performance

For the Sandy Bridge and Haswell architectures, the best performance is achieved in the 100-layer case run with 24 and 32 processes respectively (hyperthreading enabled). The results in Tables 4 and 5 show percentages of the STREAM bandwidth and the theoretical floating point throughput which incorporates the instruction balance and vectorization factors.

### 4.4 Discussion

The performance of the extruded mesh iteration is constrained by the properties of the mesh and the kernel computation. The total number of computations is based on the number of degrees of freedom per cell. The range of discretisations used in this paper (Fig. 4) leads to four cases: one, two, three or six degrees of freedom per cell. In compute bound situations, discretisations with the same number of computations have the same performance (Fig. 7).

#### 4.4.1 Temporal locality

The numbering algorithm ensures good temporal locality between vertically aligned cells. Any degrees of freedom which are shared vertically are reused when the iteration algorithm visits the next element. The reuse distance along the vertical is therefore minimal.



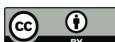

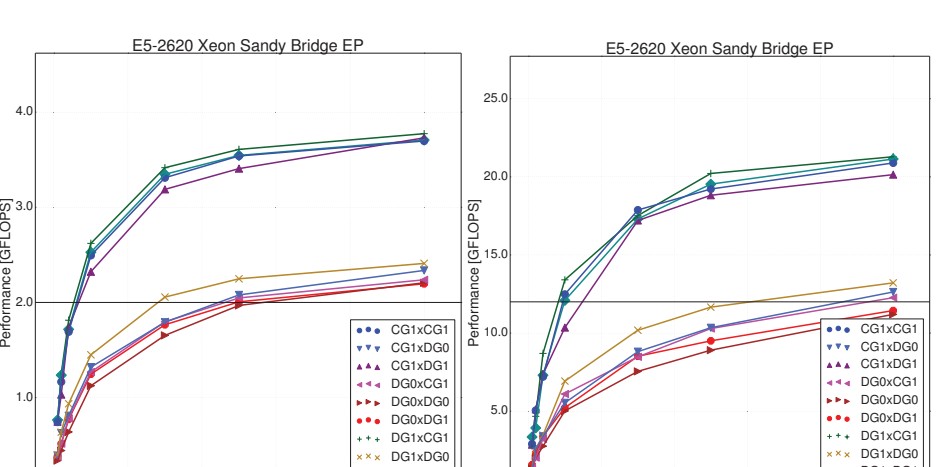

(a) Sandy Bridge, 1 process, $c = 1$      (b) Sandy Bridge, 6 processes, $c = 6$

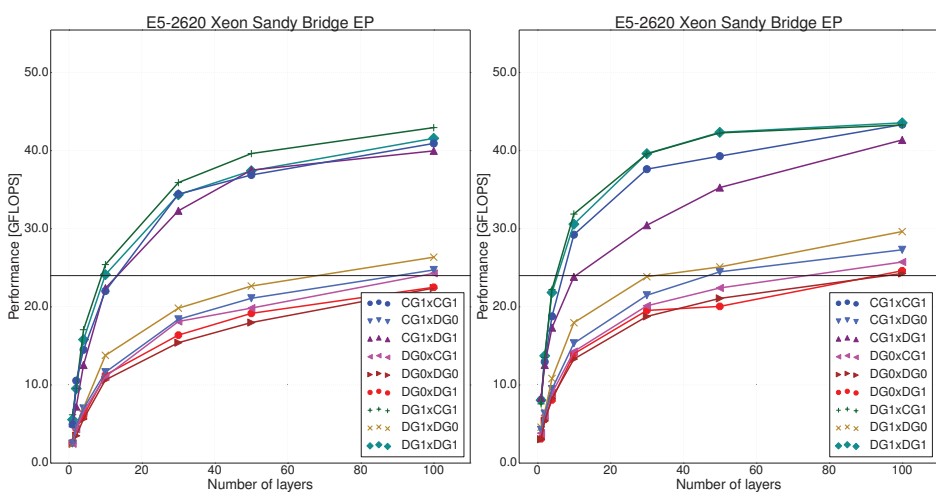

(c) Sandy Bridge, 12 processes, $c = 12$      (d) Sandy Bridge, 24 processes, $c = 12$

**Figure 5.** Performance of the $I$ integral computation with varying number of layers and number of processes on a badly-ordered base mesh. The horizontal line is the base FLOP throughput for $f_b = f_v = 1$ and the number of physical cores used.

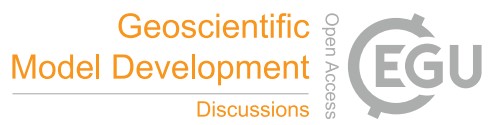



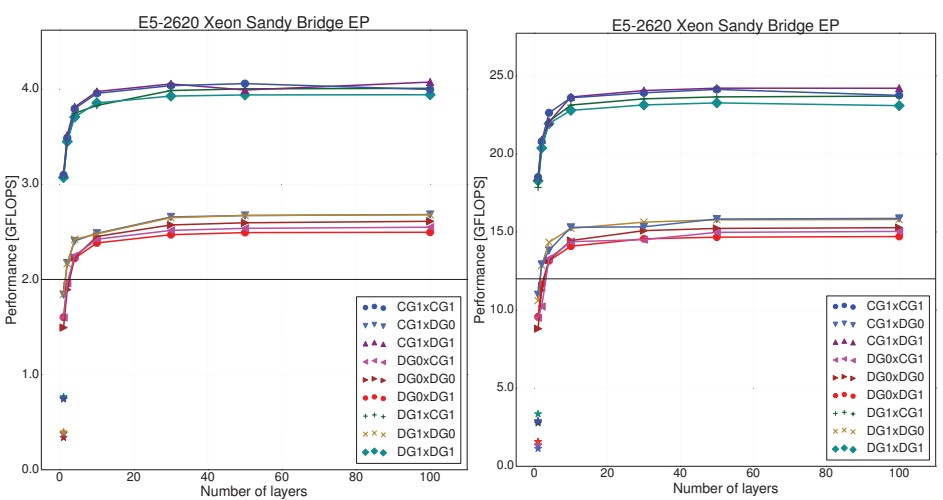

(a) Sandy Bridge, 1 process, $c = 1$        (b) Sandy Bridge, 6 processes , $c = 6$

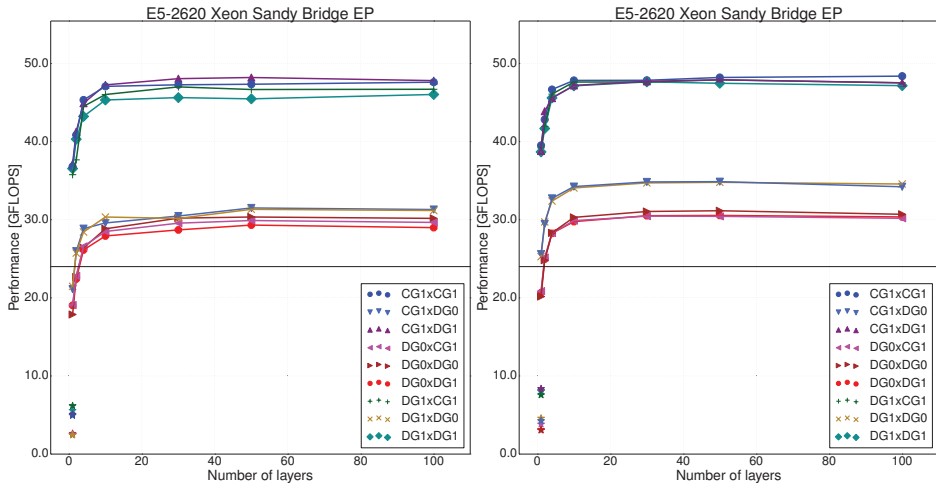

(c) Sandy Bridge, 12 processes, $c = 12$        (d) Sandy Bridge, 24 processes, $c = 12$

**Figure 6.** Performance of the $I$ integral computation with varying number of layers and number of processes. The star-shaped markers show the performance of the 1-layer badly-ordered mesh for comparison. The horizontal line is the base FLOP throughput for $f_b = f_v = 1$ and the number of physical cores used.

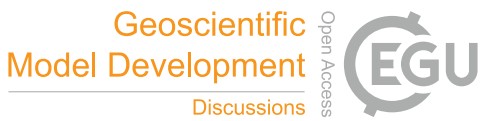



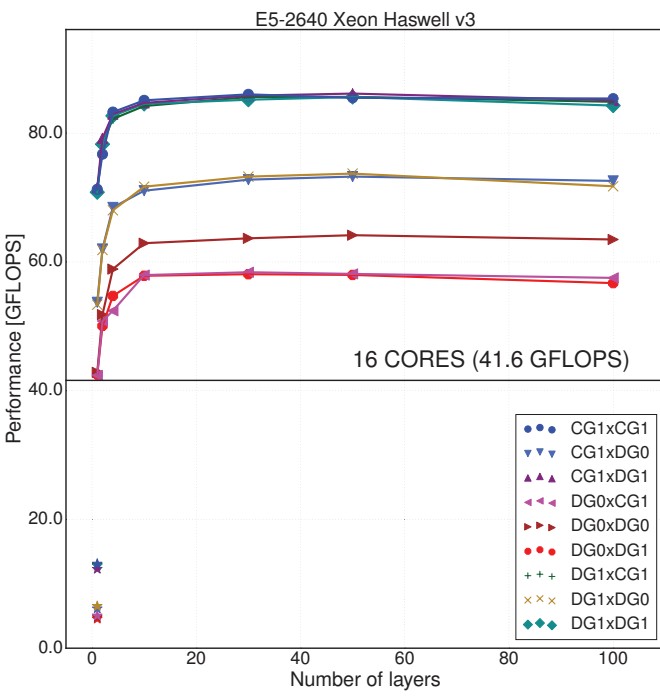

**Figure 7.** Performance of the $I$ integral computations on different data discretisations with varying number of layers on the Haswell architecture. The star-shaped markers show the performance of the 1-layer badly-ordered mesh for comparison. The horizontal line is the base FLOP throughput for $f_b = f_v = 1$ and the number of physical cores used.

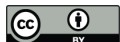



For CG discretisations, where degrees of freedom are shared horizontally with other vertical columns, the overall performance depends on the ordering of cells in the base mesh. Assuming a perfect ordering of the base mesh, the numbering algorithm ensures a minimal reuse distance while guaranteeing a minimum number of indirect accesses and satisfying all the previously introduced spatial and temporal locality requirements.

Figures 6 and 5 demonstrate the combined impact of horizontal mesh ordering and extrusion. In the extreme case the flop rate increases up to 14 times between the badly ordered single-layer case and the 100 layer well ordered case. This is consistent with the widely held belief that unstructured mesh models are an order of magnitude slower than structured mesh models.

The difference between well- and badly-ordered mesh performance outlines the benefits responsible for the boost in performance. Horizontal data reuse dominates performance for low number of layers while spatial locality and vertical temporal locality (ensured by the numbering and iteration algorithms) are responsible for most of the performance gains as the number of layers increases.

## 5 Conclusions

In this paper we have presented efficient, locality-aware algorithms for numbering and iterating over extruded meshes. For a sufficient number of layers, the cost of using an unstructured base mesh is amortized. Achieved performance ranges from 70% to 90% of our best estimate for the hardware's performance capabilities and current level of kernel optimisation. Benefits of spatial and temporal locality vary with number of layers: as the number of layers is increased the benefits of spatial locality increase while those of temporal locality decrease.

This paper employed two simplifying constraints: that there are a constant number of layers in each column, and that the number of degrees of freedom associated with each entity type is a constant. These assumptions are not fundamental to the numbering algorithm presented here, or to its performance. We intend to relax those constraints as they become important for the use cases for which Firedrake is employed.

The current code generation scheme can be extended to include inter-kernel vectorization (an optimisation mentioned in Meister and Bader (2015)) for the operations which cannot be vectorized at intra-kernel level. The efficiency of such a generic scheme applicable to different data discretisations is currently being explored.

In future work we intend to generalize some of the optimisations which extrusion enables for both residual and Jacobian assembly: inter-kernel optimisations, grouping of addition of contributions to the global system and exploiting the vertical alignment at the level of the sparse representation of the global system matrix. In addition to the CPU results presented in this paper, we also plan to explore the performance portability issues of extruded meshes on Graphical Processing Units and Intel Xeon Phi accelerators.



## 6 Team list

The Firedrake project contributors: Gheorghe-Teodor Bercea, George Boutsioukis, Colin J. Cotter, Patrick E. Farrell, Simon W. Funke, David A. Ham, Miklós Homolya, Christian Jacobs, Anna Kalogirou, Paul H.J. Kelly, Michael Lange, Nicolas Loriant, Fabio Luporini, Graham R. Markall, Andrew T.T. McRae, Lawrence Mitchell, Eike H. Müller, Florian Rathgeber, Asbjørn Nilsen Riseth, Francis P. Russell, Kaho Sato.

The Software Performance Optimisation (SPO) group members: Gheorghe-Teodor Bercea, David A. Ham, Miklós Homolya, Paul H.J. Kelly, Michael Lange, Fabio Luporini, Lawrence Mitchell, Luigi Nardi, Emanuele Rossini.

## 7 Code availability

The packages used to perform the experiments have been archived using Zenodo: Firedrake (Mitchell et al., 2016), PETSc (Smith et al., 2016), petsc4py (Dalcin et al., 2016), FIAT (Rognes et al., 2016), UFL (Alnæs et al., 2016), FFC (Logg et al., 2016), PyOP2 (Rathgeber et al., 2016) and COFFEE (Luporini et al., 2016). The source code repositories as well as the archived versions are publicly available.

## 8 Data availability

The scripts used to perform the experiments as well as the results are archived using Zenodo: Sandy Bridge (Bercea, 2016a) and Haswell (Bercea, 2016b). The archives are publicly available.

*Author contributions.* Gheorghe-Teodor Bercea designed the generalized extrusion algorithm, performed the extension of the Firedrake and PyOP2 packages to support extruded meshes, the performance evaluation and the preparation of the graphs and tables. Andrew T. T. McRae extended components of the Firedrake toolchain to support the finite element types used in the experiments, and made minor contributions to the extruded mesh iteration functionality. David A. Ham was the proponent of a generalized extrusion algorithm. Lawrence Mitchell, Florian Rathgeber and Fabio Luporini developed related features and framework improvements in Firedrake, PyOP2 and COFFEE. Luigi Nardi is responsible for the use of the floating point balance metric. David A. Ham and Paul H. J. Kelly are the principal investigators for this paper. Gheorghe-Teodor Bercea prepared the manuscript with contributions from all the authors. All authors contributed with feedback during the paper's write-up process.

*Acknowledgements.* This work was supported by an Engineering and Physical Sciences Research Council prize studentship [Ref. 1252364], the Grantham Institute and Climate-KIC, the Natural Environment Research Council [grant numbers NE/K006789/1, NE/K008951/1, and NE/M013480/1] and the Department of Computing, Imperial College London. The authors would like to thank J. (Ram) Ramanujam at Louisiana State University for the insightful discussions and feedback during the writing of this paper. We are thankful to Francis Russell at Imperial College London for the feedback on this paper.



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



**Table 1.** Topological dimensions of extruded mesh entities. $D_b$ denotes the topological dimension of the base mesh.

| Mesh entity | Dimensions |
|---|---|
| Vertex | $(0,0)$ |
| Vertical Edge | $(0,1)$ |
| Horizontal Edge | $(1,0)$ |
| Vertical Facet | $(D_b - 1, 1)$ |
| Horizontal Facet | $(D_b, 0)$ |
| Cell | $(D_b, 1)$ |



**Table 2.** Hardware used.

| Name | Intel Sandy Bridge | Intel Haswell |
| --- | --- | --- |
| Model | Xeon E5-2620 | Xeon E5-2640 v3 |
| Frequency | 2.0 GHz | 2.6 GHz |
| Sockets | 2 | 2 |
| Cores per socket | 6 | 8 |
| Bandwidth per socket | 42.6 GB/s | 56.0 GB/s |

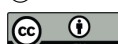



**Table 3.** Maximum STREAM triad ($a_i = b_i + \alpha c_i$) performance achieved by varying the number of MPI processes from one to twice the number of physical cores.

| Platform | STREAM bandwidth |
| --- | --- |
| Intel Sandy Bridge | 55.3 $GB/s$ |
| Intel Haswell | 80.2 $GB/s$ |

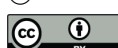



**Table 4.** Percentage of STREAM bandwidth and theoretical throughput achieved by the computation of integral $I$ over 100 layers on Sandy Bridge with 24 MPI processes.

| Discretisation | $f_b$ | $f_v$ | $P_d$ (%) | Bandwidth (%) |
|---|---|---|---|---|
| CG1 × CG1 | 1.7 | 1.58 | 73.45 | 7.092 |
| CG1 × DG0 | 1.81 | 1.0 | 78.96 | 14.70 |
| CG1 × DG1 | 1.7 | 1.58 | 73.03 | 10.50 |
| DG0 × CG1 | 1.65 | 1.0 | 76.01 | 27.86 |
| DG0 × DG0 | 1.5 | 1.0 | 85.14 | 34.86 |
| DG0 × DG1 | 1.65 | 1.0 | 75.45 | 45.68 |
| DG1 × CG1 | 1.7 | 1.58 | 73.20 | 24.60 |
| DG1 × DG0 | 1.81 | 1.0 | 78.93 | 50.98 |
| DG1 × DG1 | 1.7 | 1.58 | 71.78 | 44.37 |



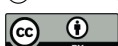

**Table 5.** Percentage of STREAM bandwidth and theoretical throughput achieved by the computation of integral $I$ over 100 layers on Haswell with 32 MPI processes.

| Discretisation | $f_b$ | $f_v$ | $P_d$ (%) | Bandwidth (%) |
|---|---|---|---|---|
| CG1 × CG1 | 1.76 | 1.61 | 72.43 | 9.015 |
| CG1 × DG0 | 1.97 | 1.0 | 88.57 | 21.92 |
| CG1 × DG1 | 1.76 | 1.61 | 72.20 | 13.39 |
| DG0 × CG1 | 1.87 | 1.0 | 73.94 | 38.74 |
| DG0 × DG0 | 1.66 | 1.0 | 91.93 | 53.10 |
| DG0 × DG1 | 1.87 | 1.0 | 72.89 | 63.11 |
| DG1 × CG1 | 1.76 | 1.61 | 71.99 | 31.19 |
| DG1 × DG0 | 1.97 | 1.0 | 87.55 | 75.17 |
| DG1 × DG1 | 1.76 | 1.61 | 71.50 | 56.98 |