# Peer review of "A numbering algorithm for finite elements on extruded meshes which avoids the unstructured mesh penalty"

_Geoscientific Model Development, 2016_

## Referee Comment (RC1) · Anonymous Referee #1 · 6 Jul 2016

The paper addresses the problem of generating sequential orders on mesh entities in so-called extruded meshes, with the goal to improve efficiency of memory access in finite element simulation software. Extruded meshes are defined as resulting from a tensor product of an unstructured mesh and a structured mesh - the paper particularly addresses layered meshes, which are unstructured 2D meshes in the horizontal and structured 1D meshes in the vertical direction (as often found in atmospheric/geoscience models). The authors present their implementation in the Firedrake software, and execute a careful analysis of achieved memory throughput for various low-order discretisation methods. Depending on the numbering of the base mesh, close-to-memory-bound performance is achieved once a certain number of layers is

exceeded.

General questions and comments:

- Data structures for layered meshes have been considered and implemented before - certainly in single-purpose codes, but also in frameworks (DUNE's prism-grid module, e.g.); I am aware that providing a survey of respective approaches to grid numbering in such packages might be impossible to do, but I think a general discussion on what options actually exist (and what implications resp. choices might have) when designing the numbering scheme could make the paper stronger.

- This is a bit related to the choice of title: at first reading I found myself expecting such a discussion; however, the paper clearly focuses on the approach followed in Firedrake (which is fine in itself, but a bit in contrast to the generic title and the abstract).

- You chose to number the DOFs of an entity contiguously, such that all DOFs of an edge (or cell) would be contiguous in memory. However, for low order methods and when the key design goal is to allow vectorization, you might want to strictly keep a stride-1 access on corresponding DOFs in layers - effectively this would mean exchanging the $l$ and $d_2$ loops in Alg. 1. In any case, this choice depends on the type of operations we expect in simulations (whether we are strongly memory or compute bound, what the memory access patterns are, etc.), so a discussion on this would be interesting.

- As far as I got it, your concept of a stencil goes beyond the strict notion typically used for finite difference methods on structured grids: your stencils may also include element-local operations in finite-element- type methods (requiring a cell and its faces, edges, vertices) or also a face-based flux operation as in finite volume methods (which might require a face and its two adjacent cells). In any

case, you might explain this in a bit more detail, and maybe state one or two examples.

• What kind of unstructured mesh did you actually use for your results? You discuss in the paper that having a structured mesh as base mesh is advantageous for performance. Hence you might even explicitly address this issue by comparing results for a structured mesh (stored in an unstructured way) and one (or more?) typical unstructured meshes from applications.

Suggestions for improving the paper:

• As my only major suggestion, I would like to encourage you to switch from GFlop/s to GB/s in all performance plots: as you are in a memory-bound regime and the numbering scheme primarily addresses achievable "valuable bandwidth", "GB/s" would be the natural metric.

• You might check whether having a log-scale for x-axes makes the results for few layers better visible

• It would be helpful to a add a sketch for illustration of the indexing scheme defined in Eq. (5)

• I was wondering what kind of stencil a DG0xDG0 discretization would produce for the residual; aren't all accesses element-local then? In general, would it make sense to add a table (or similar) that describes which entities are accessed for the various discretisations?

Typos:

• p. 5, line 17: becoems -> becomes

- in the references, line 13, it should be Günther and Pögl (with umlauts)

---

## Referee Comment (RC2) · Anonymous Referee #2 · 22 Aug 2016

**1   Context**

The article presents a numbering strategy to take advantage of the vertically structured nature of meshes used in most geophysical applications, while using a unstructured "base mesh" in the horizontal. The idea is to amortize the cost of indirect addressing in the unstructured horizontal direction through direct addressing in the vertical in the innermost loop. Although the idea is not new, as was proposed in Macdonald et al. (2011) and others, the added value is the application to a class of higher order continuous and discontinuous finite element methods A careful analysis is provided on performance for first and second order schemes, as to how many vertical levels are

required to amortize the cost of the indirect addressing.

**2   Questions and suggestions**

- I assume that $\lambda$ defined as "number of extruded layers" stands for the number of vertices of a vertical mesh, rather than the number of segments in the vertical.

  In this case, looking at Equation 5: $d_2$ can be either 0 (for vertices) or 1 (for segments). Then the statement $0 \leq l \leq \lambda - d_2$ will let $\lambda$ go out of bounds, unless the second $\leq$ symbol is replased by $<$. Similar observation for $l$-loop in Algorithm 3.

  In case this assumption is wrong, it should be made more clear what is meant with layers.

- The title suggests that besides a memory layout for function spaces, also a numbering strategy in the horizontal would be discussed.

- Looking at Figure 3., how is the numbering of $n+\#$ related to $m+\#$, and possibly nodes internal to the triangle? Are $m + \#$ and $n + \#$ related to different function spaces?

- Given the title I would have expected to see discussed what would be the impact (on e.g. performance) to number $n + \#$ (as in Figure 3) first in vertical fashion (zig-zag up-down, rather than right-left).

- It would help to see a visualisation of a practical numbering for a mesh of a few triangles and few levels, for a few function function space configurations, besides Algorithm 1.

- Algorithm 1 :
- – First "dofs$_{\text{fs}}$" is assigned with round brackets, later with square brackets.
- – The second last line makes reference to $l$, which is invalid outside the vertical loop.
- – Perhaps exchanging loops $l$ and $d_2$ can avoid the last 2 lines by looping $l$ in $\{0, 1, ..., \lambda - 1 - d_2\}$? (possibly without changing the resulting order)

- Algorithm 3 :

  - – Can I assume that for a single DG-DG cell-entity, $(\text{dof}_0, \text{dof}_1, ..., \text{dof}_{k-1})$ are contiguous?
  - – subscript fs in $L_{\text{fs}}(v)$ missing
  - – $d_2$ unassigned, should be referenced as subscript of $V$?

- Figure 5: Almost none of the results have reached as mentioned the "performance plateau" with 100 layers for the badly ordered mesh. It would be interesting to see how many layers are required for all discretizations to reach this plateau.

  Is there an expected benefit in going higher order to reach the plateau with less layers?

---

## Author Response (AR1)

**A structure-exploiting numbering algorithm for finite elements on extruded meshes, and its performance evaluation in Firedrake**

Gheorghe-Teodor Bercea1, Andrew T. T. McRae2,3,4, David A. Ham3, Lawrence Mitchell1,3, Florian Rathgeber1,5, Luigi Nardi1, Fabio Luporini1, and Paul H. J. Kelly1

1Department of Computing, Imperial College London, London, SW7 2AZ, United Kingdom
 2The Grantham Institute, Imperial College London, London, SW7 2AZ, United Kingdom
 3Department of Mathematics, Imperial College London, London, SW7 2AZ, United Kingdom
 4Department of Mathematical Sciences, University of Bath, Bath, BA2 7AY, United Kingdom
 5European Centre for Medium-Range Weather Forecasts (ECMWF), Reading, RG2 9AX, United Kingdom

Correspondence to: Gheorghe-Teodor Bercea (gb308@doc.ic.ac.uk)

**1 Responses to reviewers**

**1.1 Reviewer 1 comments**

Responses to individual comments:

**Comment 1:** Data structures for layered meshes have been considered and implemented before - certainly in single-purpose

5 codes, but also in frameworks (DUNE's prism- grid module, e.g.); I am aware that providing a survey of respective approaches to grid numbering in such packages might be impossible to do, but I think a general discussion on what options actually exist (and what implications resp. choices might have) when designing the numbering scheme could make the paper stronger.

**Answer:** We have added some additional discussion in the introduction around alternative approaches to this same problem. **Comment 2:** *This is a bit related to the choice of title: at first reading I found myself expecting such a discussion; however,*

10 the paper clearly focuses on the approach followed in Firedrake (which is fine in itself, but a bit in contrast to the generic title and the abstract).

**Answer:** The paper describes a numbering algorithm which is completely generic to finite element approaches. However, it is true that the evaluation of the approach is in Firedrake. We have therefore reworked the title to explicitly mention that the performance evaluation is in Firedrake. In addition we have added some mention of Firedrake as the tool for performance

**Comment 3:** You chose to number the DOFs of an entity contiguously, such that all DOFs of an edge (or cell) would be contiguous in memory. However, for low order methods and when the key design goal is to allow vectorization, you might want to strictly keep a stride-1 access on corresponding DOFs in layers - effectively this would mean exchanging the l and d2 loops in Alg. 1. In any case, this choice depends on the type of operations we expect in simulations (whether we are strongly memory

20 or compute bound, what the memory access patterns are, etc.), so a discussion on this would be interesting.

15 evaluation in the abstract.

**Answer:** We acknowledge that the ordering within each entity column is not unique. However, we do not see an obvious advantage to interchanging as suggested here. We have commented on this in section 3.2.

**Comment 4:** As far as I got it, your concept of a stencil goes beyond the strict notion typically used for finite difference methods on structured grids: your stencils may also include element-local operations in finite-element- type methods (requiring

5 a cell and its faces, edges, vertices) or also a face-based flux operation as in finite volume methods (which might require a face and its two adjacent cells). In any case, you might explain this in a bit more detail, and maybe state one or two examples.

Answer: The reviewer is entirely correct. We have made our definition of a stencil explicit in a new section 2.4.

**Comment 5:** What kind of unstructured mesh did you actually use for your results? You discuss in the paper that having a structured mesh as base mesh is advantageous for performance. Hence you might even explicitly address this issue by

10 comparing results for a structured mesh (stored in an unstructured way) and one (or more?) typical unstructured meshes from applications.

**Answer:** In the problem setup, we have described how we generate the base mesh, using Gmsh. Although the domain of computation is regular, the mesh itself is unstructured.

We believe the comparison to a topologically structured mesh is outwith the scope of the paper. In particular, the comments relative to structured base meshes are in place to indicate that we are explicitly depriving ourselves of these advantages, since we are aiming for an iteration algorithm that gives good performance irrespective of the base domain. Our results demonstrate that, for layered meshes, a reasonable base numbering (obtained in our case via RCM) is sufficient to obtain performance close to hardware bounds at significantly fewer layers than are scientifically interesting.

Comment 6: As my only major suggestion, I would like to encourage you to switch from GFlop/s to GB/s in all performance
20 plots: as you are in a memory-bound regime and the numbering scheme primarily addresses achievable "valuable bandwidth", "GB/s" would be the natural metric.

**Answer:** We have clarified in section 4.3 that the relevant bound is, in fact, operation count, and not bandwidth. We were ourselves surprised by this conclusion, however the performance results support this hypothesis. As such, GFlop/s is the correct metric.

Comment 7: You might check whether having a log-scale for x-axes makes the results for few layers better visible
 Answer: We tried this, but it did not create a more useful figure.
 Comment 8: It would be helpful to a add a sketch for illustration of the indexing scheme defined in Eq. (5)
 Answer: We agree and have taken the opportunity to do so as Figure 3.

Comment 9: I was wondering what kind of stencil a DG0xDG0 discretization would produce for the residual; aren't all
accesses element-local then? In general, would it make sense to add a table (or similar) that describes which entities are accessed for the various discretisations?

**Answer:** This is correct, and Figure 5 shows the degrees of freedom and therefore entities which are accessed in each case. Additionally, we have fixed the suggested typos.

**1.2 Reviewer 2 comments**

Responses to individual comments:

**Comment 1:** I assume that  $\lambda$  defined as number of extruded layers stands for the number of vertices of a vertical mesh, rather than the number of segments in the vertical. In this case, looking at Equation 5:  $d_2$  can be either 0 (for vertices) or 1

5 (for segments). Then the statement  $0 \le l \le \lambda - d_2$  will let  $\lambda$  go out of bounds, unless the second  $\le$  symbol is replaced by <. Similar observation for l-loop in Algorithm 3. In case this assumption is wrong, it should be made more clear what is meant with layers.

Answer:  $\lambda$  is in fact the number of segments in the vertical. We have explicitly defined  $\lambda$  accordingly and ensured that the usage is consistent throughout the manuscript.

10 **Comment 2:** The title suggests that besides a memory layout for function spaces, also a numbering strategy in the horizontal would be discussed.

**Answer:** We do not think that the title implies this, however, we have changed the title in response to reviewer 1. We hope that this title is less ambiguous in what it is suggesting the contributions of the paper are. We feel that the contributions of the paper are expressly stated and the title does not read contrary to those claims.

**15 Comment 3:** Looking at Figure 3., how is the numbering of n + # related to m + #, and possibly nodes internal to the triangle? Are m + # and n + # related to different function spaces?

**Answer:** The nodes internal to the triangle would be numbered in a similar way, but we feel that this would unnecessarily complicate the figure. Because the horizontal mesh is unstructured, there is no simple relationship between m and n, however our results show that using a suitable ordering of the horizontal mesh (such that m and n are typically "close") is important

20 for performance. A given function space will have degrees of freedom associated with one or more entity types, for example a continuous cubic space in the horizontal combined with a continuous linear mesh in the vertical would have the degrees of freedom shown in the figure, plus one degree of freedom per horizontal facet.

**Comment 4:** Given the title I would have expected to see discussed what would be the impact (on e.g. performance) to number n + # (as in Figure 3) first in vertical fashion (zig-zag up-down, rather than right-left).

25 Answer: Reviewer 1 also had a similar comment. We acknowledge that the ordering in each entity column is not unique. We do not see an obvious advantage to zig-zag up-down as opposed to left-right numbering. We have added some text to section 3.2 in this vein. We note that we do not believe the title claims that this is the best numbering algorithm on extruded meshes, merely an approach which works well.

**Comment 5:** It would help to see a visualisation of a practical numbering for a mesh of a few triangles and few levels, for 30 a few function function space configurations, besides Algorithm 1.

**Answer:** We do not believe this would add clarity to the paper. For example, the newly added Figure 3 merely numbers the topological entities on a single extruded triangle and is already complex.

**Comment 6:** In Algorithm 1, first "dofsfs" is assigned with round brackets, later with square brackets. **Answer:** Thanks, fixed.

**Comment 7:** *In Algorithm 1, the second last line makes reference to l, which is invalid outside the vertical loop.* **Answer:** Thanks, fixed.

**Comment 8:** In Algorithm 1, perhaps exchanging loops l and  $d_2$  can avoid the last 2 lines by looping l in  $0, 1, ..., \lambda - 1 - d_2$ ? (possibly without changing the resulting order)

5 **Answer:** Unfortunately this would change the resulting numbering.

**Comment 9:** In Algorithm 3, can I assume that for a single DG-DG cell-entity,  $(dof_0, dof_1, ..., dof_{k-1})$  are contiguous? **Answer:** Yes.

**Comment 10:** In Algorithm 3, subscript fs in Lfs(v) missing **Answer:** Thanks, fixed.

10 **Comment 11:** *In Algorithm 3, d*2 *unassigned, should be referenced as subscript of V?* **Answer:** Fixed by making *d*2 an explicit input to the algorithm.

**Comment 12:** In Figure 5, almost none of the results have reached as mentioned the "performance plateau" with 100 layers for the badly ordered mesh. It would be interesting to see how many layers are required for all discretizations to reach this plateau.

15 **Answer:** The reviewer is correct that in the badly ordered case almost none of the results have reached the performance plateau, we have reworded the discussion to address this. We do not believe that extending the experiment until we achieve some plateau in the "bad" case would be of practical interest since typical simulations have tens, not hundreds of layers in the vertical.

**Comment 13:** *Is there an expected benefit in going higher order to reach the plateau with less layers?*

20 **Answer:** There is indeed an expected benefit in going to high order, we have added a note to this effect at the end of the discussion.

**A structure-exploiting numbering algorithm for finite elements on extruded meshes<del>which avoids the unstructured mesh penalty, and</del> its performance evaluation in Firedrake**

Gheorghe-Teodor Bercea1, Andrew T. T. McRae2,3,4, David A. Ham3, Lawrence Mitchell1,3, Florian Rathgeber1,5, Luigi Nardi1, Fabio Luporini1, and Paul H. J. Kelly1 1Department 
[revised manuscript text omitted]

```

 $\begin{array}{l} (\operatorname{dof}_0,\operatorname{dof}_1,...,\operatorname{dof}_{k-1})\leftarrow L(v)\cdot(\operatorname{dof}_0,\operatorname{dof}_1,...,\operatorname{dof}_{k-1})\leftarrow L_{\operatorname{fs}}(v)\\ \text{for }l \text{ in } \{0,1,...,\lambda-d_2\} \text{ do}\\ S(f(\operatorname{dof}_0),f(\operatorname{dof}_1),...,f(\operatorname{dof}_{k-1}))\\ \text{ for }j \text{ in } \{0,1,...,k-1\} \text{ do}\\ \operatorname{dof}_j\leftarrow\operatorname{dof}_j+\operatorname{offset}_{S,\operatorname{fs}}(j)\\ \text{ end for}\\ \text{end for}\\ \text{end for} \end{array}$

**4.1.1 Choosing the computation**

Numerical computations of integrals are the core mesh iteration operation in the finite element method. We focus on residual (vector) assembly for two reasons. First, in contrast to Jacobian assembly, there are no overheads due to sparse matrix insertion; the experiment is purely a test of data access via the mesh indirections. Second, residual evaluation is the assembly operation

5 with the lowest computational intensity and therefore constitutes a worst-case scenario for data layout performance exploration. Since we are interested in data accesses, we choose the simplest non-trivial residual assembly operation:

$$I_1 = \int_{\Omega} f v \, \mathrm{d}x, \quad \forall v \in V \tag{10}$$

for f in the finite element space V. For this study we choose  $\Omega = [0, 1]^3$  to be the unit cube. The base mesh is generated in an unstructured manner using Gmsh (Geuzaine and Remacle, 2009), and then extruded to form a three-dimensional domain.

10

In addition to the output field  $I_1$  and the input field f this computation accesses the coordinate field, x. Regardless of the choice of V, we always represent x by a d-vector at each vertex of the d-dimensional mesh.

**4.1.2 Choosing the discretisations**

The construction of a wide variety of finite element spaces on extruded meshes was introduced in McRae et al. (2016). This enables us to select the horizontal and vertical data discretisations independently.

15

For the purposes of data access, the distinguishing feature of different finite element spaces is the extent to which degrees of freedom are shared between adjacent cells.